# Morganella Morganii Infection in Hirudo Medicinalis (Iran): A Case Report

**DOI:** 10.3390/vetsci9100562

**Published:** 2022-10-12

**Authors:** Hooman Rahmati Holasoo, Iradj Ashrafi Tamai, Wolfram Manuel Brück, Babak Pakbin, Alireza Nasiri, Amirparsa Azizi

**Affiliations:** 1Department of Aquatic Animal Health, Faculty of Veterinary Medicine, University of Tehran, Tehran 1419963111, Iran; 2Centre of Excellence for Warm Water Fish Health and Disease, Shahid Chamran University of Ahvaz, Ahvaz 6135783151, Iran; 3Department of Microbiology and Immunology, Faculty of Veterinary Medicine, University of Tehran, Tehran 6135783151, Iran; 4Institute for Life Technologies, University of Applied Sciences Western Switzerland Valais-Wallis, 1950 Sion, Switzerland; 5Faculty of Veterinary Medicine, University of Tehran, Tehran 1419963111, Iran

**Keywords:** *Morganella morganii*, *Hirudo medicinalis*, 16S rRNA sequencing, Antimicrobial susceptibility

## Abstract

**Simple Summary:**

*Morganella morganii* is a motile, non-spore-forming, rod-shaped facultative an-aerobic gram-negative bacterium found in the intestines of people, the oral cavity of animals, and the environment. Reptiles, guinea pigs, rabbits, jaguars, elephant seals, broiler chickens, piglets, and dolphins have all been documented to have *M. morganii* infection. Medicinal leeches are used in surgical and non-surgical manners. Treatment of long-term and chronic pain syndrome induced by degenerative diseases in a non-surgical method. For the first time in Iran, our investigation discovered *M. morganii*-infected *Hirudo medicinalis*. Infection with *M. morganii* caused a significant death and morbidity rate (70%) and severe clinical abnormalities.

**Abstract:**

Medicinal leeches (*Hirudo medicinalis*) are used in surgical and non-surgical manners. *Morganella morganii* is an opportunistic and zoonotic pathogenic bacterium causing serious clinical complications. In this study, we isolated, discovered and characterized *M. morganii*-infected *H. medicinalis*. We detected and identified *M. morganii* in all inflamed and swollen *Hirudo medicinalis* samples. The 16S rRNA sequence of the isolates confirmed all strains of *M. morganii*. All strains were sensitive to Ceftriaxone, Ceftiofur, Danofloxacin, Ciprofloxacin, Enrofloxacin, Oxytetracycline, and Meropenem and were resistant to Erythromycin, Amoxicillin, Ampicillin, Cefazolin, Colistin, Penicillin G, and Lincomycin. This pathogenic bacterium is a zoonotic pathogen, and monitoring the prevalence rate of this bacteria is strongly necessary for leeches used in human medical treatment and care. Finally, all infected leeches were treated successfully in this case report study.

## 1. Introduction

*Morganella morganii* is a motile, non-spore-forming, rod-shaped facultative an-aerobic gram-negative bacterium found in the intestines of people, the oral cavity of animals, and the environment [1]. Morgan et al. identified *M. morganii* from a pediatric fecal culture for the first time in 1906 [2]. Reptiles, guinea pigs, rabbits, jaguars, elephant seals, broiler chickens, piglets, and dolphins have all been documented to have *M. morganii* infection. *M. morganii*, for example, can live in an animal’s oral cavity and infect humans if the animal bites or scratches them. As a result, *M. morganii* is a significantly opportunistic and zoonotic pathogenic bacterium that can cause serious clinical complications. *M. morganii* infections commonly affect the urinary tract, skin and soft tissue, and the hepatobiliary tract [3]. Several virulence factors expressed in *M. morganii* contribute to this organism’s pathogenicity. Hemolysins (cytotoxic), ureases (urea as a nitrogen source, rapid urea hydrolysis, a cause of stone formation, and in-fluences bacterial growth and biofilm formation during urinary tract infections), IgA protease, and zinc metalloprotease (capable of cleaving host Ig), motility and fimbrial adhesions (uroepithelial cell adhesion and a significant determinant of bacterial colonization (protect from phagocytosis), Drug resistance genes (ampicillin, tetracycline, chloramphenicol, and sulphonamide resistance, metallo-β-lactamases resistance to tellurite). Extra genetic elements and/or mobile elements were used to confer drug resistance to *M. morganii*. The resistant strains that carry the *bla*_CTX-M_ gene can produce β-lactamases, breaking down the extended spectrum of β-lactam drugs [1].

Medicinal leeches are used in surgical and non-surgical manners. Treatment of long-term and chronic pain syndrome induced by degenerative diseases in a non-surgical method. The surgical use of leeches is to drain stagnant blood following reconstruction or plastic surgery to alleviate post-operation venous congestion [4]. Because of this treatment, more than 60% of impaired pediculated flaps and microvascular free-tissue have been saved. *H. medicinalis*, *H. orientalis*, and *H. verbena* are the three species of medical leeches. However, *H. medicinalis* is the most commonly utilized in therapeutic operations. The crop and intestine make up the leech’s digestive tract [5]. Water and salts are absorbed from the eaten, stored blood in the crop, and the ingested blood is further digested in the intestine. As digestive system flora, there are many bacterial microcolonies in the crop [6]. The use of leeches in medical and traditional medicine has increased in the last two decades due to FDA (Food and Drug Administration) approval. The breeding of this animal has increased significantly, making illness incidence in this animal a significant concern [7,8]. For the first time in Iran, our investigation discovered *M. morganii*-infected *Hirudo medicinalis*. Infection with *M. morganii* caused a considerable death and morbidity rate (70%) and severe clinical abnormalities.

## 2. Materials and Methods

### 2.1. Case and Sampling

In a leech (*Hirudo medicinalis*) production farm containing more than 30,000 leeches, mortality of leeches was observed at the rate of 10 leeches per day on averagely. All leeches were swollen and inflamed (Figure 1). Ten inflamed and swollen leech samples were collected in sterile containers and sent to the bacteriology laboratory in ice-packed coolers for further culturing and identifying probable infectious agents. After preparing smears from all specimens, they were cultured on blood agar supplemented with 5% sheep blood and MacConkey agar (Merck, Darmstadt, Germany) for 24 h at 37 degrees Celsius. All plated isolates were also given gram-stained smears. Several biochemical tests on isolates were performed to identify the bacteria, including catalase, oxidase, urease, Lysine decarboxylase, H_2_S production, and fermentation of glucose, lactose, maltose, mannitol, sucrose, and xylose (Table 1). Finally, 10 bacterial strains were isolated, identified and confirmed. All isolated strains were subjected to the further genotypic and phenotypic characterization. 

### 2.2. Antimicrobial Susceptibility Testing

The Kirby-Bauer disk diffusion method was used to determine antibiotic sensitivity. On Mueller Hinton agar (Merck, Germany), samples were evaluated for antibiotic susceptibility against standard medicine and veterinary antibiotics. All isolates were tested for resistance profile to the following antibiotics: gentamicin (GM 10 g), ampicillin (AM 10 g), penicillin G (PG 10 units), enrofloxacin (ENF 5 g), tetracycline (TE 30 g), amoxicillin (A 25 g), trimethoprim-sulfamethoxazole (SXT 25 g), erythromycin (E 15 g), ciprofloxacin (CIP 5 µg), cefoxitin (FOX 30 µg), cefazolin (CZ 30 µg), cefepime (FEP 30 µg), ceftiofur (CEF 30 µg), ceftazidime (CAZ 30 µg), ceftriaxone (CRO 30 µg), colistin (CL 10 µg), Neomycin (N 30 µg), oxytetracycline (T 30 µg), lincomycin (L 2 µg), difloxacin (DF 10 µg), danofloxacin (DFX 10 µg), meropenem (MEM 10 µg) and doxycycline (D 30 µg) according to the guidelines of the Clinical and Laboratory Standards Institute (CLSI). Following a 24-h incubation period, the results were read [9].

### 2.3. 16S rRNA Gene Amplification

Firstly, isolates of *M. morganii* were grown at 37 °C for 24 h and harvested by centrifugation at 12,000× *g* for 3 min. After being washed three times in PBS, the pelleted bacterial cells were using extraction. Genomic DNA was extracted from 48 h-old-cultures of *M. morganii* using a commercial Gram-negative bacterial DNA extraction and purification kit (Wizard^®^, Promega, Madison, WI, USA). The procedure was performed according to the manufacturer’s instructions. DNA extraction for molecular analysis was done from the bacterial cells according to the standard procedure for Gram-Negative bacteria (DNG-plusTM kit, Sinaclon, Tehran, IRAN). The extracted genomes’ quality and quantity properties were evaluated using a nanodrop spectrophotometer model ND 2000 (Thermo-Fisher Scientific Inc., Waltham, MA, USA). The 16S rRNA gene was amplified using related primers (27F, 5′-AGAGTTTGATCMTGGCTCAG-3′, and 1541R, 5′-AAGGAGGTGATCCAGCCGCA-3′). PCR mixture was prepared using 12.5 µL of 2X Master Mix (Amplicon, Odense, Danmark), 0.2 pmol of the primer (100 pmol/µL), 1 µL of template DNA (100 ng), and 9.5 µL of distilled water in a final volume of 25 µL. The reaction was carried out as follows; an initial denaturation at 95 °C for 3 min, then 34 cycles of 95 °C for 30 min, 58 °C for 30 min, 72 °C for 90 sec and a final extension at 72 °C for 7 min. The amplification products (5 µL) were resolved by electrophoresis on 1.2% agarose gel for one h at 90 V. Afterwards, the agarose gel was stained with ethidium bromide and screened using UV-illuminator. The amplified products were approximately 1500 bp [10].

### 2.4. 16S rRNA Gene Sequencing Identification

The 16S rRNA sequence coding region of strains was amplified by polymerase chain reaction (PCR). The sequences from 16S rRNA gene PCR products that were generated using universal bacterial primers 27F and 1541R were used to determine the identities of strains. Macrogene Company sequenced strains in Korea. Sanger dideoxy sequencing methods were used to obtain these sequences. The sequences derived from the isolates were analyzed using the sequences analysis by Bioedit version7 and BLAST search program (http://blast.ncbi.nlm.nih.gov/Blast.cgi), and one sequence was submitted to the Bankit. The accession number is MT840687 and the access date is 11 August 2020.

## 3. Results

Gram-negative bacilli were observed in smears obtained from the organs. After 24 h, to identify which bacteria induced the infection, we sampled specimens of the infected heart, liver, spleen, lung, and kidney to inoculate blood agar and McConkey agar in an aseptic environment. Colonies grew in MacConkey agar were lactose negative, and in Blood agar were gray, smooth, and round. The catalase, Indole, MR, motility, and urease test results were positive, and oxidase, lysine decarboxylase, voges-proskauer, and H_2_S production in TSI were negative for all isolates. The isolates non fermented lactose, sucrose, xylose and mannitol, dulcitol.

The results revealed that all isolates were sensitive to ceftriaxone, ceftiofur, danofloxacin, ciprofloxacin, enrofloxacin, oxytetracycline, and meropenem. By contrast, they were resistant to erythromycin, amoxicillin, ampicillin, cefazolin, colistin, penicillin G, and lincomycin. The susceptibility grade of the isolates against gentamycin, tetracycline, cefepime, ceftazidime, doxycycline, trimethoprim sulfamethoxazole, and neomycin were intermediate (Table 2). A DNA fragment of the expected size (1500 bp) was detected in all isolates from *Hirudo medicinalis* samples (Figure 2). The 16S-rRNA sequence of the isolated bacterial strains was compared and aligned with the known 16S-rRNA sequences of other bacteria available in the GenBank database. The isolated bacterial strain was found to have 98–100% similarity with different strains of *M. morganii*. Pool water containing the infected leeches was first renewed and then treated with the enrofloxacin antibiotic for five times at 48 h intervals through 10 days. 24 h after starting the antibiotic therapy, all clinical symptoms were reduced in the infected leeches. After 5 days of treatment, the mortality of leeches was completely prevented. 

## 4. Discussion

*Morganella* bacteria are naturally prevalent in the microbial flora in the digestive system of leeches [11]. This critter, which is utilized as a medical device, can spread this bacterium or other bacteria to humans and other animals via bite or posterior sucker, which comes from this animal’s digestive tract [12]. Due to its considerable resistance to standard medicines, this bacterium has a complicated treatment process that might result in severe death [13]. In humans, sepsis, abscessation, purple urine bag syndrome, chorioamnionitis, and cellulitis are among the illnesses linked with *M. morganii* infection. *M. morganii* causes a variety of diseases in animals, including abscessation, sepsis, and arthritis in reptiles, ocular lesions in harbour and elephant seals, bronchopneumonia in dolphins, a swollen head syndrome in broiler chickens in Japan, pleuropneumonia in piglets, pneumonia in a captive jaguar and broncho interstitial pneumonia [13,14,15]. It can also result in a high rate of mortality in chickens and cattle [3,13,16,17]. 

Several case reports have been reported, including infectious diseases caused by this bacterial pathogen in humans and animals. Abdalla et al. reported a very rare central nervous system infection with frontal brain abscess caused by *M. morganii* for the first time in a 38-year-old female patient in 2006. The treatment of antibiotic therapy was unsuccessful [18]. De et al., in another study, reported chronic osteomyelitis of the right proximal tibia caused by a biofilm-producing *M. morganii* in a 56-year-old male patient. In this case, they reported that biofilm colonization occurred on the prosthesis, fibrosed tissues, medical devices, sinus tracts, and dead bones [19]. In animal case studies, *M. morganii* infections have commonly been reported in dolphins. Between 2000 and 2019 in Australia, more than 700 infectious cases were caused by extensive antibiotic-resistant M. morganii in dolphins [20]. Recently, Han et al. in South Korea isolated the third-generation cephalosporin-resistant *M. morganii* from a captive breeding dolphin [20]. However, this bacterial pathogen has been isolated from other animals such as dairy cattle [3]. According to the writer’s knowledge, this is the first report of *M. morganii* bacteria-related sickness in leeches, which has resulted in a significant percentage of mortality in this species [10]. To prevent leech bite site infection, leech therapy was followed by lengthy antibiotic therapy [16,17]. Despite this, infection occurred most of the time, with an incidence rate ranging from 4.1 to 36.2% [8]. 

## 5. Conclusions

We isolated M. morganii from inflamed and swollen Hirudo medicinalis samples in this study. The 16S rRNA sequence of the isolates confirmed all strains of *M. morganii*. All strains were multi-drug resistant to erythromycin, amoxicillin, ampicillin, cefazolin, colistin, penicillin G, and lincomycin. Because this bacterium is a zoonotic pathogen, monitoring the prevalence rate of this bacteria is necessary for leeches used in medical treatments.

## Figures and Tables

**Figure 1 vetsci-09-00562-f001:**
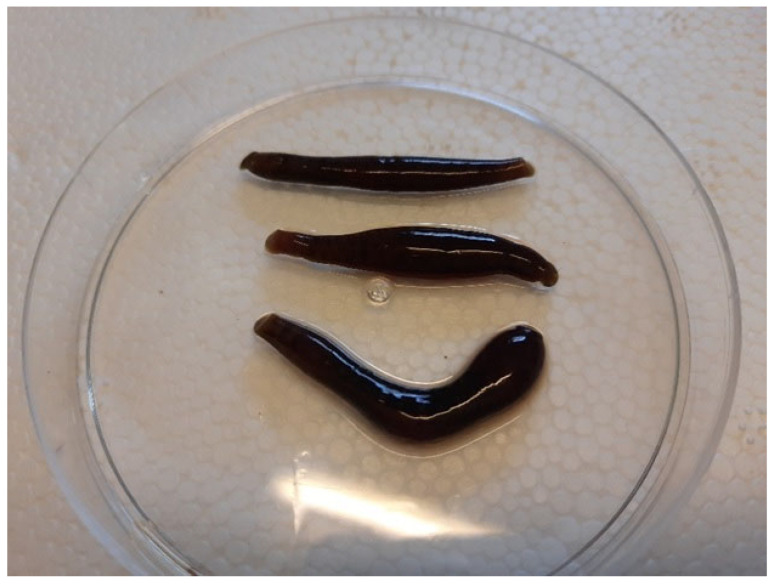
Inflamed and swollen *Hirudo medicinalis* samples.

**Figure 2 vetsci-09-00562-f002:**
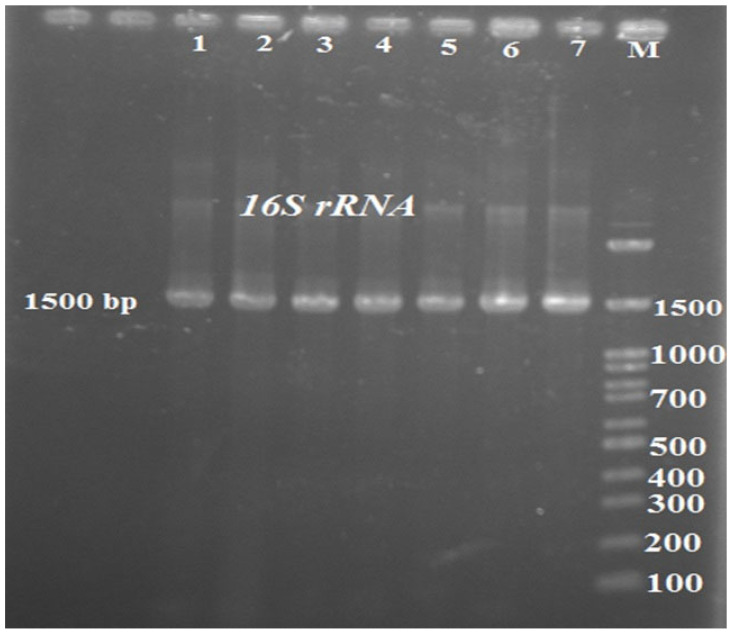
16S rRNA PCR products. (1–7: samples 1 to 7; M: marker 100 bp).

**Table 1 vetsci-09-00562-t001:** The result of Biochemical tests of *Morganella morganii*.

Biochemical Test	Results	Biochemical Test	Results
Gram staining	−	Oxidase	−
Indole	+	Catalase	+
Methyl Red	+	Trehalose	−
Voges-proskauer	−	Sucrose	−
Citrate	−	D-sorbitol	−
Urease	+	D-mannitol	−
Lysine decarboxylase	−	Dulcitol	−
Motility	+	Salicin	−
H2s Production	−	Raffinose	−

**Table 2 vetsci-09-00562-t002:** Antibiogram result of *M. morganii* isolates from *Hirudo medicinalis* samples.

Antibiotic	Sensitivity	Antibiotic	Sensitivity
A25	R	MEM10	S
D30	I	NFX5	S
CIP5	S	DFX10	S
DF10	S	AM10	R
TE30	I	PG10	R
L2	R	FOX30	I
T30	S	CZ30	R
N30	I	FEP30	I
CL10	R	E15	R
GM10	I	CEF30	S
CRO30	S	CAZ30	I
SXT	I		

S: susceptible, I: intermediate, R: resistant.

## Data Availability

All data in this study are available from the corresponding authors on a reasonable request.

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
