# Peer review of "Morganella Morganii Infection in Hirudo Medicinalis (Iran): A Case Report"

_vetsci, 2022, doi:10.3390/vetsci9100562_

Round 1

Reviewer 1 Report

Interesting work. some typo errors to fix.

Author Response

Dear Reviewer 1

Thank you very much for your valuable comment. The type errors have been fixed throughout the manuscript.

Reviewer 2 Report

In this manuscript, the authors present a case report of M. morganii infection in Hirundo medicinalis in Iran.

The authors identified and characterized strains of M. morganii isolates from several organs of H. medicinalis died in a leech production farm. The methods used are suitable for the purpose and the knowledge base regarding antibiotic sensitivities of M. morganii isolated from H. medicinalis could be useful in the context of leech therapy and could stimulate interest in the search for zoonotic bacteria and the related antimicrobial resistance in leeches used in medical treatments.

Unfortunately it is a sector of limited interest and it will be the publisher to evaluate the suitability for Veterinary Sciences journal.

As for the technical part, in my opinion the article can be published after minor changes.

Line 49-55: In this part as well as in others there are many capital initials, please check if they are really necessary.

Line 64: H. orientalis and H. verbena, change in italic, please.

Line 70: Define FDA, please

Line 197: correct “medcinalis”, please

Author Response

Dear Reviewer 2

Thank you very much for your valuable comments and they were very useful and practical to improve the quality of this research. All revisions have been considered and addressed precisely throughout the manuscript.

  1. Line 49-55: Capital initials have been revised throughout the manuscript and highlighted in the text.
  2. Line 64: H. orientalis and H. verbena, changed in italic form and highlighted in the text.
  3. Line 70: FDA is defined and highlighted in the text.
  4. Line 197: “medcinalis” is corrected and highlighted in the text.
